# Optimizing Neural Networks for Chemical Reaction Prediction: Insights from Methylene Blue Reduction Reactions

**DOI:** 10.3390/ijms25073860

**Published:** 2024-03-29

**Authors:** Ivan Malashin, Vadim Tynchenko, Andrei Gantimurov, Vladimir Nelyub, Aleksei Borodulin

**Affiliations:** Artificial Intelligence Technology Scientific and Education Center, Bauman Moscow State Technical University, 105005 Moscow, Russia; i.malashin@emtc.ru (I.M.); agantimurov@emtc.ru (A.G.); vladimir.nelub@emtc.ru (V.N.); alexey.borodulin@emtc.ru (A.B.)

**Keywords:** drug design, chemical reaction prediction, hyperparameter tuning, neural network architecture, methylene blue reduction, machine learning in chemistry

## Abstract

This paper offers a thorough investigation of hyperparameter tuning for neural network architectures using datasets encompassing various combinations of Methylene Blue (MB) Reduction by Ascorbic Acid (AA) reactions with different solvents and concentrations. The aim is to predict coefficients of decay plots for MB absorbance, shedding light on the complex dynamics of chemical reactions. Our findings reveal that the optimal model, determined through our investigation, consists of five hidden layers, each with sixteen neurons and employing the Swish activation function. This model yields an NMSE of 0.05, 0.03, and 0.04 for predicting the coefficients A, B, and C, respectively, in the exponential decay equation *A* + *B* · *e*^−*x*/*C*^. These findings contribute to the realm of drug design based on machine learning, providing valuable insights into optimizing chemical reaction predictions.

## 1. Introduction

The kinetic assessment of chemical reactions is pivotal in elucidating reaction mechanisms and optimizing processes across diverse domains of chemistry. Among these, the reduction of Methylene Blue (MB) by ascorbic acid (AA) [1,2] stands as a notable reaction, owing to its widespread applications in analytical and industrial contexts, attributed to its sensitivity and specificity. MB is a water-soluble cationic dye molecule [3], readily reduced to leucomethylene blue, and oxidized back to its blue form. The reduction reaction of MB is actively researched by scientists worldwide [1,4,5], reflecting its significance in various fields of study. Despite challenges in employing it as a clock reaction, its kinetics remain a valuable teaching tool, measured spectrophotometrically at λ=665 nm, offering insights into rate law dependencies on MB, ascorbic acid, and other concentrations.

The reduction of MB has been scrutinized and extensively discussed in academic literature. The optimal conditions for decolorizing methylene blue using ascorbic acid were determined by Weiland et al. [6] by investigating the wavelength and intensity of illumination. Apples and potatoes were halved, immersed in a methylene blue solution, and exposed to light. Decolorization was observed, revealing the tissue’s reducing capacity, with apples showing greatest effect around the core and under the skin, while potatoes exhibited less decolorization, primarily in a band near the skin.

Seno et al. [7] proposed that the reduction of MB with L-ascorbic acid in aqueous surfactant solutions is accelerated by hexadecyltrimethylammonium bromide (HTAB) [8] but not by tetrabutylammonium bromide. Micellar effects, affected by pH and inhibited by KCl, suggest changes in substrate dissociation and product binding to micelles.

MB reduction to leucomethylene blue (MBH) by ascorbic acid in acetonitrile is studied by Hallock et al. [9] using cavity ring-down spectroscopy (CRDS) [10]. This technique allows for precise monitoring of concentration changes on a microsecond timescale, which was previously unattainable. CRDS utilizes reflective mirrors to form an optical cavity, and changes in light intensity are recorded as the absorber affects the decay time. This method offers significantly increased sensitivity compared to traditional absorption measurements.

The impact of the anionic surfactant sodium dodecyl sulphate (SDS) pre-micellar clusters on the electron transfer reaction between MB and AA in dilute acid conditions was explored by Sen et al. [11]. The reaction, exhibiting first-order kinetics with respect to both MB and AA, involves uncatalyzed and *H*^+^-catalyzed paths. Iodide ion was found to accelerate the reaction rate.

Simple classroom experiments, like clock reactions, offer accessible avenues to grasp advanced concepts. For instance, Snehalatha et al. [12] demonstrated MB and L-ascorbic acid clock reaction to undergraduates, showcasing kinetics principles with visible color changes. This approach contrasts with the classic “Blue Bottle” experiment and has been expanded to include studies on persulfate-iodide and bromination reactions, elucidating factors like ionic strength and substituent effects on reaction rates.

Recent literature has seen a surge in studies investigating the properties and behaviors of MB through the lens of machine learning (ML) algorithms. Kooh et al. [13] employ supervised ML algorithms to model MB dye adsorption by Azolla [14] pinnata, aiming for accurate predictions of adsorption capacity across various conditions. SVR-RBF [15] emerges as the top-performing algorithm, achieving an R value of 0.994 with minimal error.

As global regulations tighten on water discharge color, the need for affordable dye removal solutions grows. Marzban et al. [16] investigates MB removal using sodium alginate-kaolin beads, assessing various parameters’ impact on efficiency. Characterization and modeling techniques reveal an artificial neural network (ANN) [17] as the most accurate predictor. The developed model aids in optimizing treatment processes, achieving efficient removal of basic dyes. Adsorption kinetics [18] followed a pseudo-second-order model, and encapsulating kaolin in sodium alginate significantly enhanced removal efficiency.

Asfaram et al. [19] investigates zinc sulfide nanoparticles with activated carbon (ZnSNPs-AC) for MB adsorption. Characterization methods and modeling techniques, including RSM [20], ANN, and least squares-support vector machine (LS-SVM), were employed. LS-SVM [21] and ANN models outperformed central composite design (CCD) [22] in predicting MB adsorption efficiency.

Investigations into the adsorptive removal of MB dye from water using different parts of an Abelmoschus esculentus (lady’s finger) [23] with processed seed powder (LFSP) are described by Nayak et al. [24]. Batch studies evaluated biosorption performance under varying conditions, revealing pseudo-second-order kinetics and intra-particle diffusion as significant factors. Langmuir and Temkin isotherms provided the best fit. Thermodynamic analysis indicated spontaneous and endothermic processes. RSM combined with ANN showed RSM’s dominance in both data fitting and estimation capabilities.

Water contamination from artificial dyes in domestic and industrial wastewater poses a significant environmental challenge, necessitating effective adsorbents for dye removal. Soleimani et al. [25] addresses water contamination by utilizing calcium alginate hydrogels reinforced with cellulose nanocrystals (CA/CNC) [25] as an effective and environmentally friendly adsorbent for MB removal. Characterization confirmed CA/CNC’s suitability, with adsorption kinetics following the pseudo-first-order model and equilibrium data fitting the Langmuir isotherm model. RSM and an artificial neural network integrated with the whale optimization algorithm (ANN-WOA) [26] were applied for analysis, with ANN-WOA demonstrating superior predictive accuracy. These findings highlight CA/CNC hydrogels as promising bio-adsorbents for addressing MB contamination in water.

Mahmoodi et al. [27] introduces a novel approach by coupling an artificial neural network (ANN) with particle swarm optimization (PSO [28] to predict MB adsorption by a starch-based superadsorbent. The superadsorbent, synthesized with acrylic acid and acrylamide polymers, was functionalized with catecholamine groups via dopamine polymerization. Comparing the performance, ANN-PSO showed superior accuracy compared to RSM, and the prediction model for adsorption capacity was exemplified through a set of explicit equations arranged in a matrix format.

This paper introduces a novel approach by coupling an artificial neural network (ANN) with particle swarm optimization (PSO) to predict methylene blue (MB) adsorption by a starch-based superadsorbent. The superadsorbent, synthesized with acrylic acid and acrylamide polymers, was functionalized with catecholamine groups via dopamine polymerization. ANN-PSO outperformed response surface methodology (RSM) in accuracy, with the best ANN topology identified as 3-7-1. The ANN-PSO model yielded lower root-mean-square error (22.46), higher correlation coefficient (0.99), and normalized standard deviation (16.83) compared to RSM (82.89, 0.98, and 65.41, respectively).

Alibak et al. [29] employ cascade correlation neural network (CCNN) [30] modeling to simulate the removal of MB using a bio-based hydrogel. The CCNN, trained with the Levenberg-Marquardt algorithm, predicts MB uptake based on factors such as adsorbent type, temperature, initial dye concentration, pH, and contact time. Results show excellent agreement with experimental data, confirming that MB uptake intensifies with temperature, contact time, pH, and initial dye concentration, while the hydrogel reinforced with bentonite/GO nanoparticles proves to be the most effective adsorbent.

In the landscape of chemistry and engineering, leveraging machine learning (ML) to forecast the coefficients of the rate law governing the interactions involving MB marks a significant advancement. Our study undertakes a thorough examination of the kinetic characteristics associated with the reduction of MB by ascorbic acid within the super diluted solutions (SDSs) of MB, water, and ascorbic acid at various concentrations. By scrutinizing the reaction kinetics across these diverse solvent matrices, our aim is to elucidate the nuanced correlation between solvent composition and reaction rate, ultimately identifying the most sustainable model for predicting rate law coefficients through neural network hyperparameters tuning [31].

## 2. Results

### 2.1. Spectroscopic Analysis Results

Based on the conducted measurements, a total of 220 files were obtained, each containing at least 10 spectra corresponding to different combinations of reactions with varying concentrations of mixed substances. Figure 1 illustrates examples of absorption intensity spectra over time for various reactions with differing concentrations.

### 2.2. Exponential Fitting

As previously mentioned, these spectra were fitted with an exponential function. In some cases, the fitting process failed due to the limited number of measurements, resulting in the library’s inability to capture the decaying trend accurately. Such profiles were excluded from further analysis.

Consequently, after filtering, a dataset for training was obtained, consisting of 180 records of conducted reactions. In some instances, the same reaction was conducted multiple times (specifically, six times). Examples of such reactions are illustrated in Figure 2.

### 2.3. Hyperparameters Optimization Pipeline

To enhance the efficiency of predicting coefficients A, B, and C in decay profiles, a Python code was developed to fine-tune the architecture [32] and the hyperparameters of a neural network model tailored for regression tasks. The grid search method explores a range of parameters to optimize the neural network architecture. These parameters include the number of hidden layers [33] (1 to 5), the number of neurons per layer [34] (16 to 256), activation functions (ReLU [35], Sigmoid [36], Tanh [37], Leaky ReLU [38], ELU [39], and Swish [40]), training epochs (50 or 100), and batch sizes [41] (16 or 32). Each combination of these parameters is tested to identify the optimal configuration for the neural network model.

For each parameter combination, a neural network model is instantiated with the specified architecture and hyperparameters. The model is trained on a training dataset and evaluated on a separate test dataset. During evaluation, the Normalized Mean Squared Error (NMSE) [42,43] is calculated for each target variable (A, B, C) to assess the model’s predictive performance.

Subsequently, the NMSE scores obtained for each parameter combination are stored, along with the corresponding architecture and hyperparameters. These results are then saved in an Excel file, with columns representing the architecture and hyperparameters used for training, as well as the NMSE scores for each target variable.

Throughout this process, the grid search [44] seeks to identify the optimal combination of model architecture and hyperparameters that minimize prediction error for the given regression task. This systematic approach facilitates the exploration of a wide range of model configurations, leading to informed decisions regarding the most effective neural network architecture for the predictive task at hand. Figure 3 illustrates the results of NMSE scores for predicting the coefficients A, B, and C.

In the course of our study, where we systematically explored parameter combinations using a grid search on our dataset, we identified optimal models with the lowest normalized mean squared error (NMSE) values. The following models demonstrated the best performance:A model with 1 hidden layer containing 256 neurons, utilizing the ELU activation function, and trained for 100 epochs. For this model, the NMSE values for coefficients A, B, and C were 0.011391, 0.019457, and 0.124704, respectively.A model with 1 hidden layer containing 64 neurons, utilizing the ELU activation function, and trained for 100 epochs. For this model, the NMSE values for coefficients A, B, and C were 0.152417, 0.083142, and 0.110623, respectively.A model with 5 hidden layers containing 16 neurons each, utilizing the Swish activation function, and trained for 100 epochs. For this model, the NMSE values for coefficients A, B, and C were 0.053186, 0.033048, and 0.037663, respectively.

All these models were trained using a batch size of 32 and 100 epochs. The performance of each model was evaluated using cross-validation, resulting in variance in the NMSE metrics. Other models exhibiting low NMSE are presented in Table 1.

## 3. Discussion

### 3.1. Optimization of Neural Network Hyperparameters for Chemical Reaction Prediction

The optimization of hyperparameters for neural network architectures is critical in achieving optimal performance for datasets related to chemical reactions, which encompass information on various volumetric compositions of reacting substances. This study underscores the importance of carefully selecting and fine-tuning the architectural parameters of neural networks to accurately model and predict the behavior of complex chemical systems. By systematically exploring different combinations of layers, neurons, activation functions, epochs, and batch sizes through techniques like grid search, we can identify configurations that yield the lowest errors and the highest predictive accuracy.

Recent advances in artificial neural networks, particularly in deep learning, have demonstrated their efficacy in drug discovery and toxicology research. The study [45] explores various hyper-parameter configurations for deep neural networks and finds that optimized models outperform traditional cheminformatics methods, with non-linear models performing well under low noise levels and Naïve Bayes models excelling at higher noise levels.

Computational simulations offer promise in rational material design, but their high computational cost limits their ability to efficiently explore material phase space. The study [46] employs machine learning to predict a critical parameter in catalyst performance, the reaction barrier, advancing the computational prediction of catalytic activity.

A demonstration of the effectiveness of transformer-based models in automatically inferring reaction classes from simple text-based representations of chemical reactions is shown in [47]. Additionally, the authors showed that the learned representations serve as reaction fingerprints, capturing nuanced differences between reaction classes better than traditional methods. These findings facilitate a deeper understanding of chemical reaction spaces, as illustrated by an interactive reaction atlas offering visual clustering and similarity searching capabilities.

### 3.2. Exploring Architectural Choices and Model Performance

The results obtained from this optimization process provide valuable insights into the interplay between architectural choices and model performance [48]. For instance, we observed that certain configurations, such as networks with a higher number of layers and neurons, tend to capture more intricate patterns in the data but may also be prone to overfitting, especially with limited training data. On the other hand, simpler architectures with fewer layers and neurons may generalize better to unseen data but could potentially overlook subtle nuances in the dataset.

Furthermore, the choice of activation functions plays a crucial role in shaping the non-linear relationships between input features and output predictions [49]. By experimenting with a variety of activation functions like ReLU, sigmoid, tanh, leaky ReLU, ELU, and swish, researchers can gain deeper insights into how different activation functions influence the learning dynamics and convergence of neural networks.

Additionally, the impact of training parameters such as the number of epochs and batch size on model convergence and computational efficiency cannot be understated. Fine-tuning these parameters based on the dataset size, complexity, and computational resources available is essential to strike a balance between training time and model performance.

Overall, the findings from this study pave the way for future research endeavors aimed at further refining neural network architectures for chemical reaction datasets. Potential avenues for future exploration include the investigation of more advanced neural network architectures such as recurrent neural networks (RNN) [50] or attention mechanisms, ensemble methods for combining multiple neural network models, and the incorporation of domain-specific knowledge into model design through techniques like transfer learning or neural architecture search. Moreover, exploring the applicability of neural networks in conjunction with other machine learning techniques like support vector machines (SVM) [51] or random forests (RF) [52] could offer complementary insights and enhance predictive capabilities for chemical reaction datasets.

## 4. Materials and Methods

### 4.1. Preparation of Chemical Reactions

The study focuses on investigating the kinetic properties of the chemical reaction involving the reduction of methylene blue by ascorbic acid in various solvent systems. These solvent systems encompass super dilute solutions (SDS) of methylene blue, ascorbic acid, and water, as well as purified water. Rigorous characterization procedures will be employed to ensure the uniformity and purity of each solvent system prior to experimentation. Subsequently, four distinct reactions will be conducted as follows:MB (dissolved in $H_{2}O$) + AA (dissolved in $H_{2}O$) → ProductsMB (dissolved in a SDS of $H_{2}O$) + AA (dissolved in a SDS of $H_{2}O$) → ProductsMB (dissolved in a SDS of MB) + AA (dissolved in a SDS of AA) → ProductsMB (dissolved in a SDS of AA) + AA (dissolved in a SDS of MB) → Products

The subsequent task entailed assessing the kinetic characteristics (reaction rate of neutralization or decolorization) of the model reaction upon mixing its components with varying concentrations of super-dilute solutions (SDSs). To achieve this, methylene blue was pre-mixed with the following:SDS MB/SDS water/water (in a volume ratio of 1:1)SDS MB/SDS water/water (in a volume ratio of 9:1)SDS MB/SDS water/water (in a volume ratio of 99:1)

A solution of MB is prepared in the designated solvent and transferred into a cuvette with a volume of 2 mL. The cuvette is then placed in the cuvette compartment of the spectrophotometer, and the absorption spectrum of the solution is recorded in the range of 500 to 700 nm. The measurements obtained on the spectrophotometer PB 2201 (SOLAR, Belarus, Minsk ) involved recording at least 10 consecutive readings with a 30 s interval between each. Subsequently, the maximum absorbance for each of the curves was identified, resulting in first-order decay plots for MB+ absorbance at 665 nm.

### 4.2. Processing of Results

A Python script was developed to automate the processing of data obtained from the spectrophotometer. This script facilitated the conversion of cumbersome .txt files into more manageable .xlsx files, each storing spectral curves at various time points, with a new curve generated every 30 s. The script also automatically detected the maximum of each spectrum, typically located around 665 nm as theorized. Upon detection, it created .xlsx files for the decay of the maximum absorbance. Using the lmfit library, the script then performed curve fitting using an exponential decay function of the form
(1)$$A+B\cdot e^{-x/C}$$

The resulting fits were visualized alongside the original data, providing insight into the decay behavior. Consequently, a dataset was compiled containing filenames corresponding to reaction experiments and coefficients obtained from curve fitting.

The dataset was constructed by recording the volumes of substances involved in each reaction, where each row represented a distinct reaction scenario. For instance, if the initial mixing volume for substances was 2000 mL and the volumetric ratios of MB and AA were 99:1, then 99 and 1 were entered into the respective columns for MB and AA. Similarly, if SDS of AA and H_2_O were added in a 9:1 ratio, 9 and 1 were recorded in the corresponding AA_SDS and H_2_O columns. Additionally, an extra solution was consistently introduced in a volume of 120 mL. Up to six substances could be included: MB, SDS MB, AA, SDS AA, H_2_O, and SDS H_2_O. An attempt was made to incorporate temperature data during solution measurements and to indicate the influence of sunlight exposure, but these additions did not contribute to model improvement. Further columns, denoted as MB+, AA+, MB_SDS+, AA_SDS+, H_2_O+, and H_2_O_SDS+, were appended to include additional substances.

Figure 4 illustrates the proposed neural network architecture for predicting coefficients A, B, and C in chemical reactions. The diagram showcases the input parameters of the model, allowing for variation in the number of neurons and hidden layers. Despite these variations, the model focuses on predicting the three coefficients, providing a streamlined approach to reaction prediction.

## 5. Conclusions

In this study, we investigated the optimization of hyperparameters for the architecture of a neural network for analyzing data related to chemical reactions. Our dataset contains information on various volumetric compositions of substances involved in the reactions. The primary aim of our investigation was to ascertain the optimal parameters of the neural network for accurately predicting the profiles of exponential decay coefficients in chemical reactions.

In the process of hyperparameter optimization, we used the grid search method, exploring different combinations of parameters such as the number of layers, the number of neurons per layer, activation functions, the number of training epochs, and batch size. This approach allowed us to systematically investigate various configurations of neural networks and evaluate their performance on our dataset.

The results of our study showed that the choice of optimal hyperparameters significantly influences the performance of the neural network. Our investigation unveiled the significant impact of activation functions on enhancing the efficiency of the neural network. We found that for certain types of data and tasks, specific activation functions may be more suitable than others. For example, the Swish function performed best for certain types of chemical data, while ReLU was preferable for others.

Overall, our study confirmed the importance of selecting the right hyperparameters to ensure the optimal performance of neural networks in analyzing chemical data. Our findings can be valuable for researchers and engineers working in the fields of chemistry and machine learning and can serve as a basis for further research in this area.

## Figures and Tables

**Figure 1 ijms-25-03860-f001:**
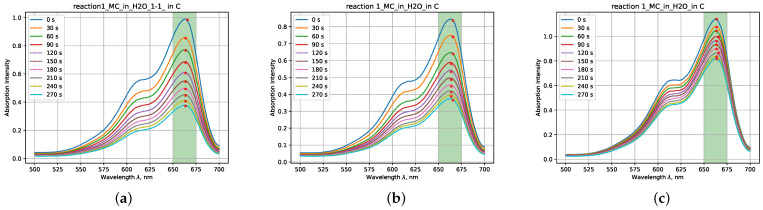
Examples of absorbtion intensity spectra for reaction MB+H_2_O (**a**–**c**) with varying volumetric ratios of mixed substances at 1:1, 9:1, and 99:1, respectively.

**Figure 2 ijms-25-03860-f002:**
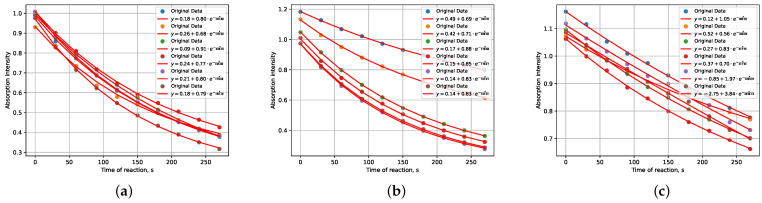
Examples of combined decay profiles with original data of maximum absorbtion intensity for reactions (**a**–**c**) MB+H_2_O, conducted six times with varying volumetric ratios of mixed substances at 1:1, 9:1, and 99:1, respectively. (each point represents the maximum absorbance intensity at λ=665 nm).

**Figure 3 ijms-25-03860-f003:**
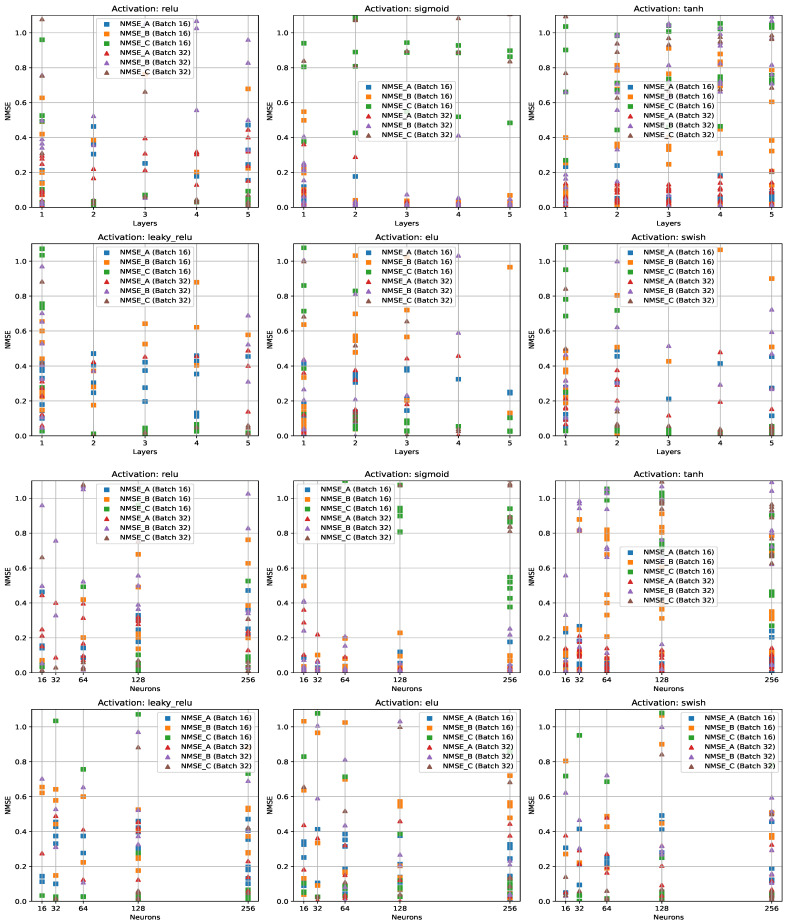
NMSE scores representing coefficients A, B, and C in approximating decay intensity profiles, varying across neural network hyperparameters, are visualized in the plane of NMSE against the number of layers and neurons.

**Figure 4 ijms-25-03860-f004:**
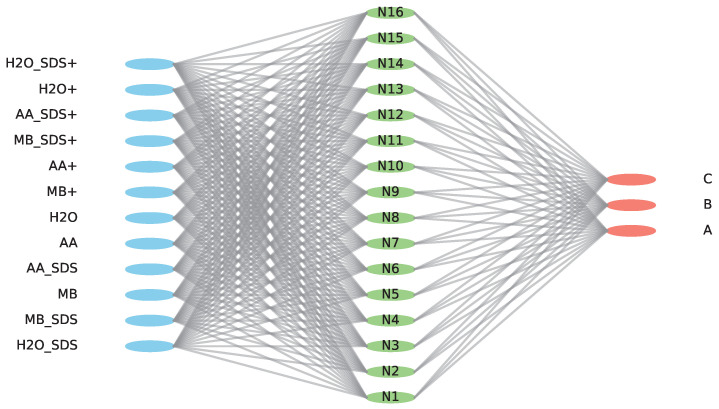
One of the possible Neural Network Architectures for Predicting Coefficients A, B, and C: a variant with sixteen neurons and one hidden layer.

**Table 1 ijms-25-03860-t001:** Best Models Performance Summary.

Hidden Layers	Neurons	Activation	Epochs	Batch Size	NMSE_A	NMSE_B	NMSE_C
1	256	ELU	100	16	0.01 ± 0.005	0.02 ± 0.008	0.12 ± 0.03
2	64	ELU	100	32	0.15 ± 0.02	0.08 ± 0.01	0.11 ± 0.02
2	128	Leaky ReLU	100	16	0.25 ± 0.03	0.18 ± 0.02	0.01 ± 0.005
2	256	ELU	100	32	0.14 ± 0.02	0.21 ± 0.03	0.09 ± 0.015
3	128	ELU	100	16	0.21 ± 0.02	0.20 ± 0.03	0.03 ± 0.01
4	128	ReLU	50	16	0.18 ± 0.02	0.20 ± 0.03	0.03 ± 0.01
5	16	Swish	100	32	0.05 ± 0.01	0.03 ± 0.005	0.04 ± 0.008
5	128	ReLU	100	16	0.25 ± 0.03	0.22 ± 0.02	0.06 ± 0.01

## Data Availability

The code and datasets for this research are openly accessible on our Git repository: ChemReactOpt, https://github.com/catauggie/ChemReactOpt, accessed on 24 March 2024. This centralized resource promotes transparency and facilitates reproducibility in scientific research. We invite the scientific community to explore, scrutinize, and build upon our work, fostering collective progress in the field.

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
