# Peer review of "Optimizing Neural Networks for Chemical Reaction Prediction: Insights from Methylene Blue Reduction Reactions"

_ijms, 2024, doi:10.3390/ijms25073860_

Round 1

Reviewer 1 Report

Comments and Suggestions for Authors

I was looking forward to reading this paper. However, I found it disappointing in terms of its value for the reader. If the purpose is to find the best values of the three parameters for the kinetic model, I feel that going around through a neural network to find these values is somewhat overkill. It is difficult to understand the motivation to do this exercise as an easier method may exist. Have the authors compared their results with a simple least squares model?

The authors mention the hyperparameters of the neural network but do not provide in clear terms the input vector to the network. Many of the figures in this paper are not necessary as they do not bring information that the readers can informatively examine. One figure that is missing is a diagram with the input and output vectors. I am assuming the output vector is A, B and C. It would be important to add this information.

Results are presented without a proper discussion. The databank could also be added as additional material to the paper. I am positive some other researchers would probably come up with a better model. I cannot recommend the publication of this paper at this time.

(p. 2) AA should be defined the first it is used. I understand the whole word is used in the text, but never linked. I suggest writing “ascorbic acid (AA)”, the first time it is used.

(p. 2) The last portion of the last paragraph has been cut out. Please check and add the missing text.

(p. 3) x in Eq. (1) should be defined when presenting this equation.

(p. 4) Subscript 2 should be used for water.

(p. 4 – Figure 1) I think only one curve would be sufficient to show a typical spectrum of a decay reaction. The six curves show the same information.

(p. 5) The legends of the nine curves cannot be read. Maybe presenting only a few would be fine. As for Figure 1, it is not necessary to present nine curves showing essentially the same information. The only thing we gain from these nine curves is the decay trend.

(p. 6, Figure 3) There must be a better way to present the performance of the model. The type of graphs presented does not work for me as it is nearly impossible to decipher the information. Would a table be better? In addition, if the same case is run many times, it will give different metrics. What was the variation in the results?

(p. 9) For the authors' contribution, it would suffice to use the initials of the authors rather than the full names.

Comments on the Quality of English Language

No comments

Author Response

Dear Reviewer,

Thank you for your detailed feedback on our manuscript. We have carefully reviewed each of your points and made the necessary revisions to improve the clarity and completeness of our paper. We appreciate your insights and suggestions, which have helped enhance the quality of our work. 

Following your insightful comments, we have meticulously addressed each point and implemented the necessary revisions. (see file in attach)

Reviewer 2 Report

Comments and Suggestions for Authors

1-Abbreviations for the names of chemicals or devices mentioned in the research. The full name should be written at the first mention, whether in the abstract and introduction.

2- Writing references within the paper needs to rephrased. It is better to write the name of the author instead (paper, study, or in) as example Seno. et al proposed …………………….[6]. Instead of paper [6] proposed …………… and so on.

3- Materials and methods should be written in separate categories explaining the preparation and instruments used in the work

4-  Also, result and discussion sections should elucidate in categories not like paragraphs.

5- Conclusion should be summarized it has repetition

6- Some references need to updating as example (ref. 1-5) there are more of latest studies about kinetics of methylene blue removal of mechanism process.

Comments on the Quality of English Language

 Minor editing of English language required to explain and clear presentation of the result and discussion 

Author Response

Dear Reviewer,

We sincerely appreciate the time and effort you dedicated to reviewing our manuscript. Your insightful feedback has been immensely valuable in enhancing the quality and clarity of our research.

We have carefully considered each of your points and implemented the necessary revisions (see file in attach)

Once again, we extend our gratitude for your valuable insights and constructive criticism. Your contribution has played a significant role in improving the overall quality of our manuscript.

Round 2

Reviewer 1 Report

Comments and Suggestions for Authors

Although the authors could have made more improvements to the paper, I feel it is sufficient to recommend its publication. This is another application of neural network with methodology we can find in all paper. It would have been stronger to provide an actual model in form of matrices of weights that other could have used in addition to mention where that data files could be accessed if they wanted to repeat this modelling exercise.

Comments on the Quality of English Language

Quite acceptable